# Nerve Sparing Radical Hysterectomy: Short-Term Oncologic, Surgical, and Functional Outcomes

**DOI:** 10.3390/cancers12020483

**Published:** 2020-02-19

**Authors:** Mustafa Zelal Muallem, Robert Armbrust, Jörg Neymeyer, Andrea Miranda, Jumana Muallem

**Affiliations:** 1Department of Gynecology with Center for Oncological Surgery, Charité–Universitätsmedizin Berlin, Corporate Member of Freie Universität Berlin, Humboldt-Universität zu Berlin, and Berlin Institute of Health, Virchow Campus Clinic, Charité Medical University, 13353 Berlin, Germany; Robert.armbrust@charite.de (R.A.); Andrea.miranda@charite.de (A.M.);; 2Department of urology, Mitte Campus Clinic, Charité Medical University, 10117 Berlin, Germany; joerg.neymeyer@charite.de

**Keywords:** nerve-sparing radical hysterectomy, inferior hypogastric plexus, pelvic splanchnic nerves, hypogastric nerves, parametrium, paracolpium

## Abstract

There is an obvious prevalence of disparity in opinions concerning the technique of nerve-sparing radical hysterectomy and its application, despite agreement on the need to spare the pelvic autonomic nerve system during such a radical operation. Understanding the precise three-dimensional anatomy of paracolpium and its close anatomical relationship to the components of the pelvic autonomic nervous system is the key in performing the nerve-sparing radical hysterectomy. A total of 42 consecutive patients with primary cervical cancers, who were operated upon in our institution between January 2017 and June 2019, were analyzed, concerning surgical, urinary functional, and short-term oncologic outcomes. Two thirds of the patients had locally advanced tumors (T > 40 mm or pT ≥ IIA2) with a median tumor size of 44.1 mm. The nerve-sparing radical hysterectomy was combined with the complete recovery of bladder function in 90% of patients directly after surgery and in 97% of patients in the first 2 weeks. The recurrence rate in a median follow-up time of 18 months was 9.5%. The nerve-sparing radical hysterectomy approach, which depends on the comprehensive understanding of the precise entire anatomy of paracolpium, was found to be feasible and applicable, even in locally advanced tumors, with good functional results and convincing short-term oncologic outcomes.

## 1. Introduction

In their latest review (2017) [1] on Querleu–Morrow classification of radical hysterectomy (2008) [2], Querleu et al. emphasize the appropriation of the nerve-sparing technique to radical hysterectomy type C adapted to the International Federation of Gynecology and Obstetrics (FIGO) stage IB1 (the old version from 2008 [3]) with deep stromal invasion and IB2-IIA or early IIB cervical cancers. This has not been defined arbitrarily, but rather because large and/or deep-infiltrating tumors will not be sufficiently operated on without a total resection of the vesicouterine (ventral parametrium) and vesicovaginal ligaments (ventral paracolpium) combined with a resection of adjusted length of the vaginal vault and its surrounding paracolpium. This radical resection (type C) sacrifices the ventral part of the inferior hypogastric plexus, the bladder branches, and probably the pelvic splanchnic nerves (leading to type C2) without prior direct visualization and dissection of the bladder branches and inferior hypogastric plexus. Some investigators, who claimed that nerve-sparing radical hysterectomy has to be restricted only to small tumors in early-stage cervical cancers [4,5,6,7], ignored that the fact that technique was designed to deal with vesicovaginal ligament (ventral paracolpium) with sparing the bladder branches of inferior hypogastric plexus. Resecting of ventral paracolpium (vesicovaginal ligament) is necessary only in more advanced tumors [8].

In this study, we describe our technique for nerve-sparing radical hysterectomy. This technique incorporates illustrating all parts of the pelvic autonomic nervous system and transecting only the uterine branches in order to preserve the hypogastric nerves, the pelvic splanchnic nerves, and the bladder branches of the inferior hypogastric plexus. We also report the urologic functional and short-term oncologic outcomes of this technique.

## 2. Materials and Methods

Forty-two patients with primary cervical cancer who underwent nerve-sparing radical hysterectomy (type C1) at our institution between January 2017 and June 2019 were included in this analysis. All data were documented in a validated data bank. Informed consent was obtained before collecting clinical data. Cancer staging in the study was adjusted to The International Federation of Gynecology and Obstetrics (FIGO) classification from 2019 [9]. Patients were thoroughly counseled about the different possible choices of treatment. In particular, in cases greater than or equal to stage IB3 disease, they were informed of concomitant chemo-radiation as standard treatment, and that radical hysterectomy represented here an experimental option. Institutional Review Board approval was obtained under registration number EA1/174/14.

The laparoscopic approach was the standard of care in our institution until the publication of Laparoscopic Approach to Cervical Cancer (LACC) Trial [10]. Afterwards, we changed our standard approach to open surgery. All nerve-sparing radical hysterectomies were performed as standardized by the first author (M.Z.M.). During nerve-sparing radical hysterectomy, individual blood vessels and nerve bundles were meticulously identified and separated. Three of the included patients were operated with live video transmission to a conference audience (two patients during the 9th and 10th International Charite Mayo conferences, Berlin, 2017 and 2019, and one patient during W.A.L.L.S. (Women Abdominal Laparoscopic/Laparotomic Live Surgery), Palermo, 2018) [11].

For visual illustration of the autonomic nervous system, we used ultrasonic liposuction SonoSurg (Olympus Deutschland GmbH, Hamburg, Germany). During surgical procedures, pelvic intraoperative neuromonitoring (pIOM) was used to support nerve-sparing radical hysterectomy and to confirm the intraoperative functional integrity of the inferior hypogastric plexus. The modular technology consisted of an ISIS Xpert neuromonitoring system including a neurostimulator and multichannel amplifiers for electromyographic recording (EMG). The NeuroExplorer as operating software included the pIOM application, providing simultaneous stimulation and recording with the help of a hand-guided disposable 400 mm ball-tip stimulation probe and needle electrodes (inomed Medizintechnik GmbH, Emmendingen, Germany). The needle electrodes were placed in the internal and the external sphincter.

All patients received intraoperatively a bladder catheter with connected pressure sensor for neuromonitoring. The catheter was removed on the third day after laparoscopic nerve-sparing radical hysterectomy and on the fifth day after open surgery. Directly after the surgery, all patients received MYOCHOLINE-GLENWOOD 10 mg, three times a day for 3 weeks. After removing the bladder catheter, the pre-voiding urine volume at the first desire to void and the post-voiding residual urine volume was measured three times a day using the ultra-sound examination for the next 3 days before discharging and once again after 3 and 6 months. The primary endpoint for bladder function (integrity of parasympathetic innervation) was the post-voiding residual urine volume of less than 100 mL on the day of catheter removal and less than 50 mL 2 days later, as well as 3 and 6 months postoperatively. Secondary endpoint of bladder function (integrity of sympathetic innervation = sensation of bladder fullness) was the pre-voiding urine volume at the first desire to void between 200 and 400 mL measured by ultrasound. This study was instituted according to a template for intervention description and replication (TIDieR) checklist and guide [12]. The study is a retrospective analysis of prospective collected database.

### Surgical Steps for Nerve-Sparing Radical Hysterectomy

After preparing the paravesical and pararectal space, the lateral parametrium (uterine artery and vein and their accompanying lymph tissue), the dorsal parametrium (sacrouterine ligament), and the ventral parametrium (vesicouterine ligament) were identified (Figure 1).

Subsequently, the hypogastric nerves bilaterally and the superior hypogastric plexus at the lateral side of rectum about 2–3 cm caudal from the ureter were identified (Figure 2).

After isolating and lateralizing the hypogastric nerves bilaterally, the dorsal parametrium (sacrouterine ligament) and the dorsal paracolpium (sacrovaginal ligament) to the tendinous arch of the pelvic fascia (fascia pelvis visceralis) were resected (Figure 3).

The uterine artery and vein (lateral parametrium) were resected at the internal iliac artery and vein and prepared to the middle point above the ureter (Figure 4, resection at the left side).

The ventral parametrium was prepared by entering the ureter tunnel medial from the ureter and rolling the ureter itself laterally and ventrally to the pelvic sidewall and pubic symphysis. Here, a small artery arising from the uterine artery and ending in the ureter (the ureteral branch of uterine artery) and a small vein going from the ureter to the uterine vein (ureteral vein) were identified. In 50% of cases, medial from these two small vessels, an arterial branch of uterine artery was identified. This branch crossovers the proximal ureter and goes along the lateral vaginal wall. It is, in our experience, the vaginal branch of the uterine artery. After isolation and cutting all these three branches, the resection of the ventral parametrium was completed and the ureter was then completely unroofed (Figure 5).

At the internal iliac vessels, two groups of vessels: the inferior vesical artery and vein (the last medial branch of internal iliac vessels), the vaginal vein (often wrongly called as deep uterine vein), and in 50% of cases the vaginal artery were identified (Figure 6a,b).

By cutting the vaginal vessels (lateral paracolpium) only and directly at the level of internal iliac vessels, the pelvic splanchnic nerves were revealed. These nerves ran directly from the sacral roots in front of the common trunk of the internal pudendal and inferior gluteal vessels at the dorsal edge of the lesser sciatic foramen, and then medial from the inferior vesical vein cranially to merge in the inferior hypogastric plexus behind the ascending vaginal vessels (Figure 7).

The vaginal vein showed two vein anastomoses with the inferior vesical vein. We called these two connected veins the lateral vesicovaginal vein and the medial vesicovaginal vein. These two veins with the accompanying lymph tissue build the vesicovaginal ligament (ventral paracolpium). Isolating these two veins and resecting them directly beneath the bladder revealed the bladder branches of the inferior hypogastric plexus (Figure 8).

Mediodorsal from the bladder branches, lateral from the vaginal wall, and above the level of hypogastric nerve, a small space (hollow) can be developed, which we called the Fujii space, honoring Prof. Shingo Fujii who described it for the first time [13]. This space allowed us to isolate the inferior hypogastric plexus from the lateral vaginal wall and to isolate uterine branches of inferior hypogastric plexus to be resected from the pelvic autonomic nervous system (Figure 9).

At this point, the uterus was only connected to the vagina. The vaginal vessels go along the vaginal wall, medial from the inferior hypogastric plexus, and could be cut and ligated at the adjusted length of vaginal vault appropriate to the tumor size and infiltration (Figure 10).

## 3. Results

Table 1 and Table 2 summarize the characteristics of patients, tumors, and operations.

The median age of our study collective was 45.26 years (29–69). Pre-operative magnetic resonance imaging was performed in 83.3% of patients and revealed suspected pelvic lymph nodes in 16.7% of cases and suspected infiltration of parametrium in 23.8%.

A total of 57% of patients had locally advanced disease (38% FIGO Ib3 and 19% FIGO ≥ IIA2) in the pre-operative clinical evaluation. The median tumor size was 44.1 mm (12–100 mm). In total, 78.6% of tumors were squamous cell cancers, and 50% of tumors were poorly differentiated whereas the other 50% were moderate differentiated tumors.

In 40 out of 42 patients, the nerve-sparing radical hysterectomy could be performed bilaterally. In two patients, we were forced to cut the inferior hypogastric plexus at one side because of the tumor infiltration into the paracolpium.

A total of 57% of nerve-sparing surgeries were performed laparoscopically and 43% as open surgery. There were no cases of conversion from laparoscopy to open surgery. The median tumor size in laparoscopic cases was 47.6 mm and the median tumor size in open surgery cases was 39.4 mm.

The resection of only sentinel lymph nodes, which is our standard for tumors smaller than 20 mm and if the bilateral sentinel lymph nodes are tumor free, was performed in two cases (4.8%). Pelvic lymph node dissection was performed in 59.5% of cases and pelvic and paraaortal lymph node dissection was performed in 35.7% (our standard indicates the pelvic and paraaortic lymph node dissection if the sentinel lymph node is positive or if we have suspected lymph nodes in pre-operative imaging). The median number of resected paraaortal lymph nodes was 16.7 (5–60) and the median number of harvested pelvic lymph nodes was 44.9 (12–89). A total of 9.5% of patients had affected paraaortal lymph nodes and 47.6% had positive pelvic lymph nodes. The infiltration of parametrium/paracolpium was the case in 52.4% of patients, and the pathological invasion in the vagina was diagnosed in 26.2% of cases. The TNM classification revealed that two thirds of patients had locally advanced tumor (T > 40 mm or pT ≥ IIA2). In 54.8% of the cases, the tumor invaded the lymph vascular space. The resection in microscopic free margins was achieved in all cases.

The median operation time was 280 min (275.5 min in the laparoscopic cases and 286.6 min in the open surgery cases).

The median duration of hospital stay was 10.8 days. The median blood loss was 97.7 mL.

There were 10 postoperative complications in grade II-IIIa according to Clavien–Dindo classification [14]. Two patients in this study suffered from localized ureter ischemia (at the most distal part of the ureter) 14 and 19 days after operation. Both patients were treated with endoscopic trans-cystic insertion of Allium Ureteral Stent (Allium Medical, Caesarea, Israel). Both cases were diagnosed after laparoscopic approaches and in the early stage of this study when we resected the inferior vesical vessels during the resection of lateral and ventral paracolpium.

Further complications included three patients with urinary tract infections, one patient with sub-ileus, three patients with lymphatic oedema, and one patient with wound dehiscence.

In 24 cases, our tumor board advised an adjuvant therapy; in 21 cases (50% of all patients), concurrent radiochemotherapy; in 2 patients, only adjuvant radiotherapy; and in 1 patient, only adjuvant chemotherapy. Five patients did not follow the recommendation of adding adjuvant therapy.

### 3.1. Functional Results

Figure 11 shows the pelvic neuromonioring intra-operatively after nerve sparing radical hysterectomy in one of our study patients.

In 38 patients (90.5%), the bladder catheter was able to be removed on the third or the fifth post-operative day according to protocol. In four patients (9.5%), three after laparoscopic approach and one after open surgery, we had to insert the bladder catheter again and had to start the bladder training because of the high post-voiding residual urine and/or the impairment sensation of bladder fullness. Three out of these four patients finally achieved the primary and secondary endpoint for bladder function within the first 2 weeks after surgery. The last patient with unilateral resection of inferior hypogastric plexus was able to void sufficiently in the first 6 weeks after surgery, but her impaired sensation of bladder fullness got worse after adjuvant radiochemotherapy, and thus she needed to empty her bladder regularly every 3 to 4 hours without waiting for the fullness sensation.

The rate of nerve-sparing failure in our collective was as low as 2.4%, in spite of the high-risk cases with more than 50% of parametrium/paracolpium infiltrations and adjuvant therapy.

### 3.2. Short-Term Oncologic Outcomes

After a median follow-up of 18 months (5–35 months), there were four recurrences (9.5%): one in the paraaortal region, two at the pelvic sidewall, and one inguinal relapse. All these patients were nodal positive patients and all of them received adjuvant radiochemotherapy. In all these recurrences the nerve-sparing surgery was performed laparoscopically.

## 4. Discussion

In our previous paper, we emphasized upon restricting the term nerve-sparing radical hysterectomy only to surgical techniques that include a clear description of the surgical steps required to dissect and spare all components of the pelvic autonomic nervous system (hypogastric nerves, pelvic splanchnic nerves, and the bladder branches of the inferior hypogastric plexus) [8]. Fujii et al. described in 2007 the precise anatomy of the vesicouterine ligament for radical hysterectomy and showed for the first time the meticulous separation of the blood vessels and connective tissues to preserve the pelvic splanchnic nerve, the hypogastric nerve, and the bladder branch of the inferior hypogastric plexus under magnification [15].

Our technique, described in this study, differed from the Fujii technique in these essential ways:We identified the entire course of pelvic splanchnic nerves from the sacral roots to their insertion in the inferior hypogastric plexus.We stopped to sacrifice the inferior vesical vessels, which would otherwise be isolated, clamped, cut, and ligated according to the Fujii technique [15], after our two complications with distal ureter ischemia. Inferior vesical vessels proved to be the most reliable anatomical landmarks to identify the pelvic splanchnic nerves in our technique.We recognized and described the entire three-dimensional anatomy of paracolpium and the essential role of vaginal vessels in the anatomy of radical hysterectomy. This allowed us to resect a quite long vaginal vault (for large tumor) without injuring the components of the inferior hypogastric plexus and therewith to operate upon the locally advanced tumor in a nerve-sparing context.

This technique was able to achieve the complete recovery of bladder function in 90% of patients directly after surgery and in 97% of them in the first 2 weeks. These good functional results did not impair in the 38% of patients who received additional radio- or radiochemotherapy, perhaps because we were able to achieve free margins in the entire patient collective, with which the adjuvant radiotherapy was applied at only 45 Gy and did not include a vaginal brachytherapy boost.

A recurrence rate of less than 10% after a median follow-up of 18 months with locally advanced tumors in two thirds of the cases may be considered as a good result in comparison with the results of definitive primary radiochemotherapy for the same collective of patients [16,17]. Some authors might argue that the primary radiochemotherapy was the standard therapy in two thirds of our patient group, and thus the combined modality treatment should be avoided [18,19]. It worth stating that there is no controlled randomized study to support this. There is a single randomized trial comparing surgery with or without postoperative radiotherapy versus definitive radiotherapy alone in patients with FIGO stage IB or IIA [20]. This study was not designed for locally advanced tumors and was not powered to answer the question about combined therapy in a high-risk group. Of all controlled randomized studies in high-risk cervical cancers, the GOG-109 trial that added radiochemotherapy after radical hysterectomy yielded the best survival results [21]. Our new understanding of the three dimensional paracolpium and its precise anatomy may be the best platform for adjusting the radicality of this operation. In our opinion, in early-stage cervical cancer without vaginal infiltration and therewith no need for a big vaginal vault (FIGO IA-IB1), it would be acceptable to spare the vaginal vessels (wrongly referred to as deep uterine vein) and their anastomoses to inferior vesical vessels.

Concerning on our two localized ischemic ureter necrosis, Wertheim reported that 6.4% of his patients developed ureterovaginal fistulae following the ureter necrosis [22]. He proposed that preservation of the vessels leading to the ureters is only of minor importance; it is more important to preserve the sheath of the ureter [23]. However, we believe that the inferior vesical artery plays a very important role in the blood supply of the most distal part of ureter and a special effort has to be made to spare these vessels during the radical hysterectomy, especially by resecting the ventral paracolpium. Our conviction falls in tandem with the anatomical work from Fröber [24] and the results of Triboulet and Vandenbussche’s study [25]. After studying 74 sections of female ureters, Triboulet and Vandenbussche concluded that there are two different anastomotic vascular systems along the wall of the ureter. In the presence of an arterial arcade (in 76% to 88% of cases) the likelihood of ischaemia is smaller. If this arcade is replaced by fine vessels such as a mesh net in a plexiform system, necrosis can only be avoided if the arterial supply is preserved and a careful dissection of the ureter has been carried out. The strengths of our study came from using a standardized operation technique with very clearly described surgical steps, the clearly defined functional tests and avoiding the surgical bias by performing all operations by one surgeon. A limitation of our study is the retrospective character of the analysis of a prospective collected database and the short follow-up period.

## 5. Conclusions

In conclusion, our nerve-sparing radical hysterectomy approach was found to be feasible and applicable even in locally advanced tumors, with good functional results and 90% disease-free survival at 18 months of follow-up. These promising outcomes suggest more research and longer follow-up to be able to verify our initial results before being recommended as a reasonable alternative to primary radiochemotherapy as standard of care for locally advanced cervical tumors.

This paper is the first that describes a precise three-dimensional anatomy of paracolpium and its close anatomical relationship to the components of the pelvic autonomic nervous system. This may be the key for the right adjusting of radicality in cancers limited to the cervix only.

## Figures and Tables

**Figure 1 cancers-12-00483-f001:**
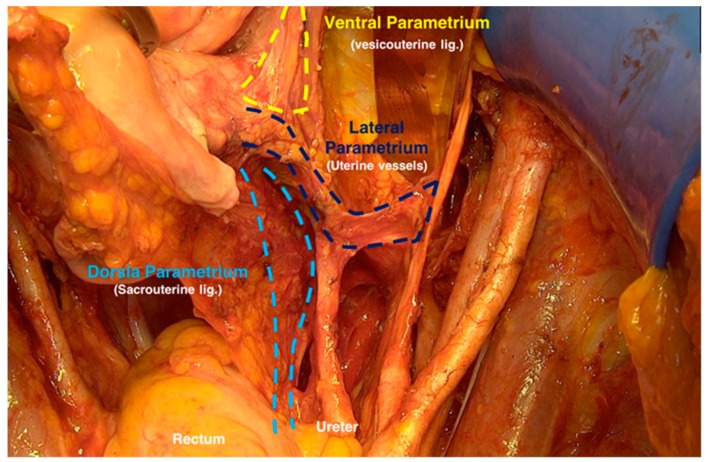
Identifying the ventral, lateral, and dorsal parametrium after preparing the paravesical and pararectal spaces.

**Figure 2 cancers-12-00483-f002:**
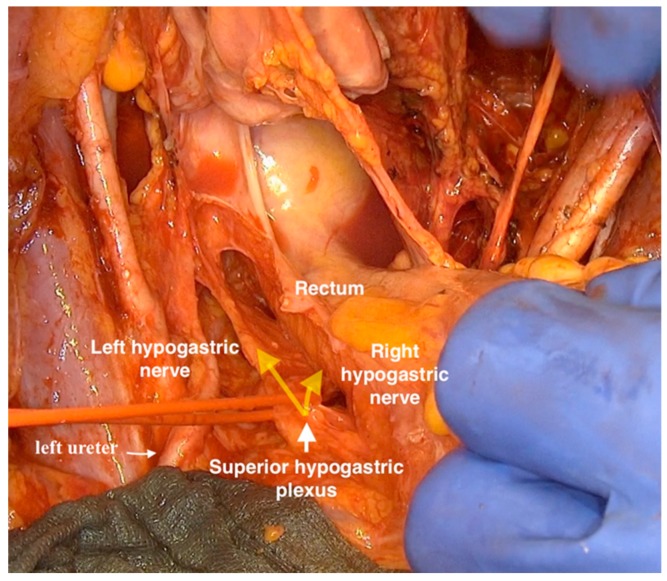
Highlighting the hypogastric nerves and superior hypogastric plexus.

**Figure 3 cancers-12-00483-f003:**
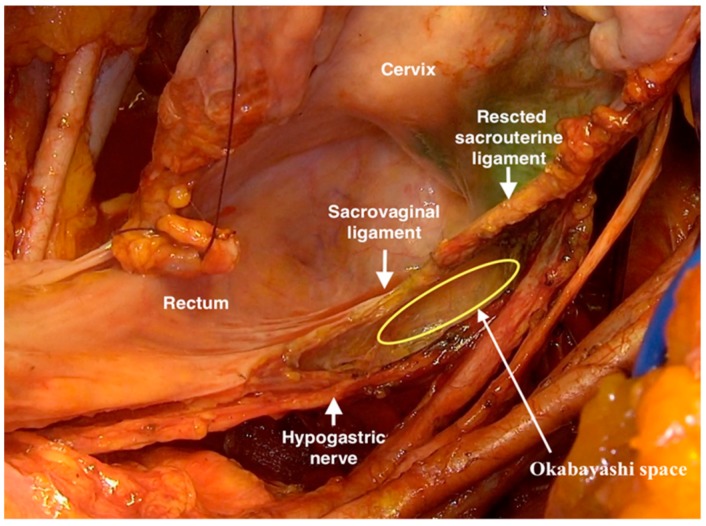
Resection of the dorsal parametrium (sacrouterine ligament) and the dorsal paracolpium (sacrovaginal ligament) to the tendinous arch of the pelvic fascia with sparing the hypogastric nerves and the dorsal part of inferior hypogastric plexus.

**Figure 4 cancers-12-00483-f004:**
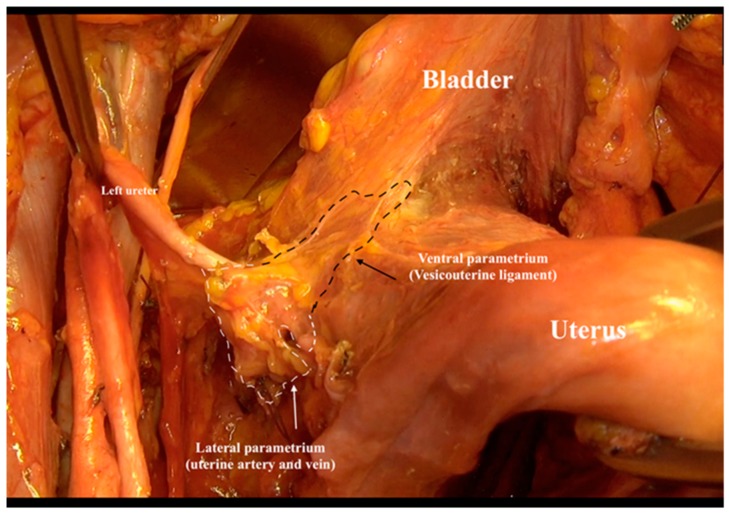
Resection of the uterine artery and vein (lateral parametrium) at the internal iliac artery and vein.

**Figure 5 cancers-12-00483-f005:**
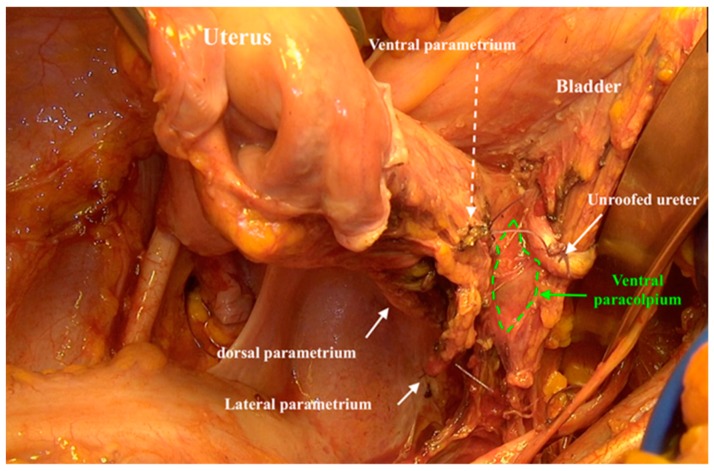
The cut-ends of ventral, lateral, and dorsal parametrium and illustrating the entire three-dimentional anatomy of paracolpium.

**Figure 6 cancers-12-00483-f006:**
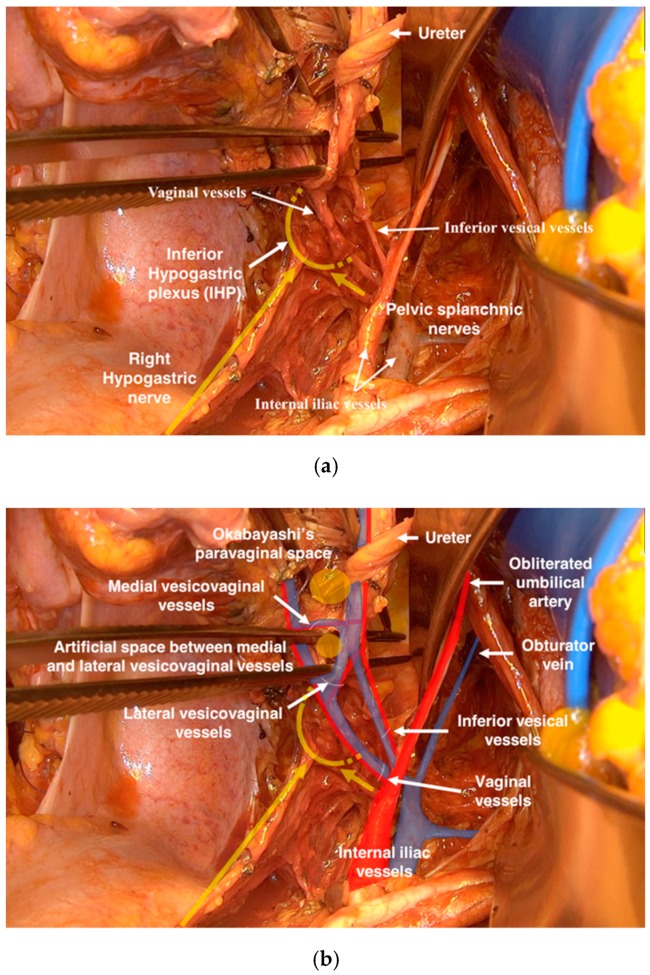
(**a**) The adherent localization between lateral and ventral paracolpium and the pelvic splanchnic nerves and inferior hypogastric plexus. (**b**) The vascular anastomoses between vaginal vessels and inferior vesical vessels in the ventral paracolpium.

**Figure 7 cancers-12-00483-f007:**
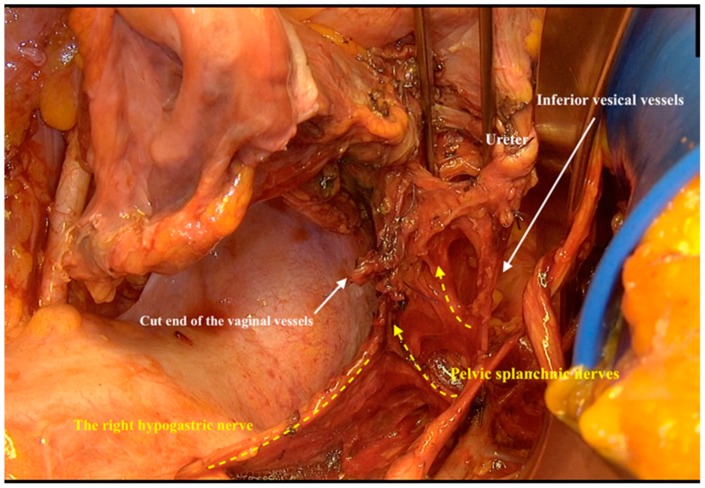
Exposing the inferior hypogastric plexus after resection of dorsal and lateral paracolpium.

**Figure 8 cancers-12-00483-f008:**
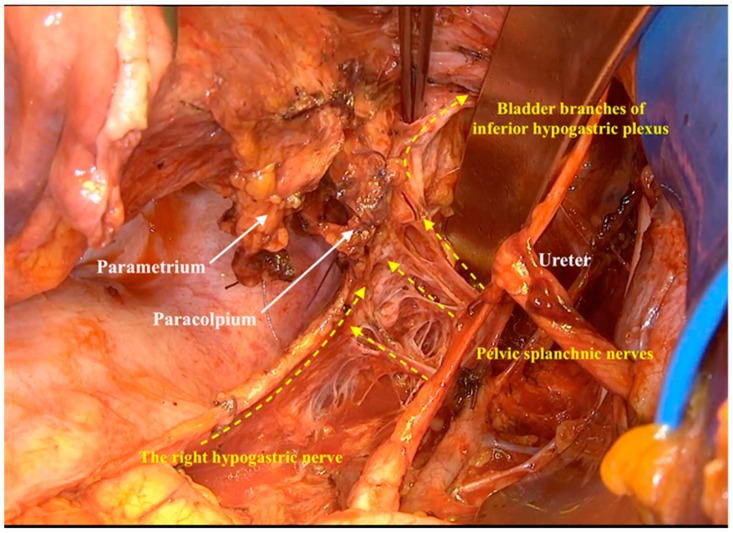
Highlighting all the components of inferior hypogastric plexus after resecting the ventral paracolpium.

**Figure 9 cancers-12-00483-f009:**
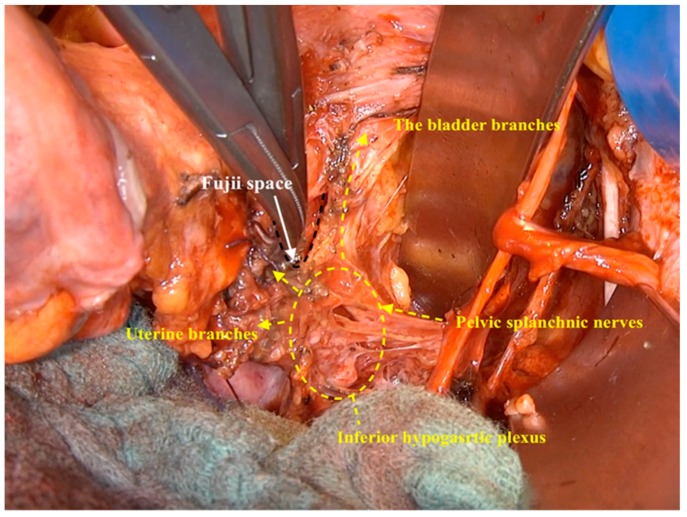
The dissection of the Fujii space to isolate the inferior hypogastric plexus from the lateral vaginal wall.

**Figure 10 cancers-12-00483-f010:**
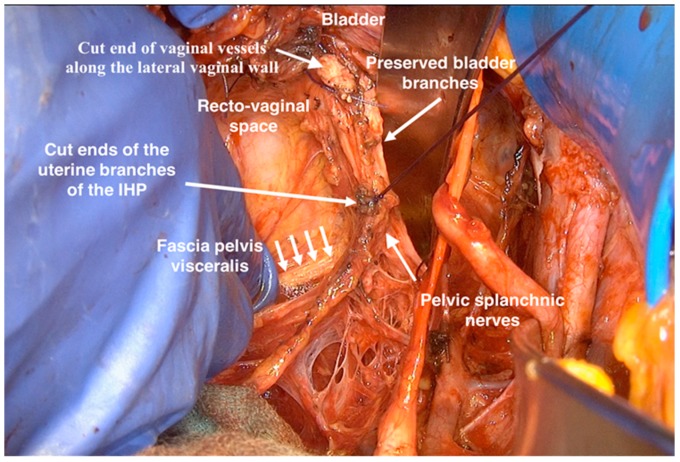
The spared components of inferior hypogastric plexus after radical hysterectomy.

**Figure 11 cancers-12-00483-f011:**
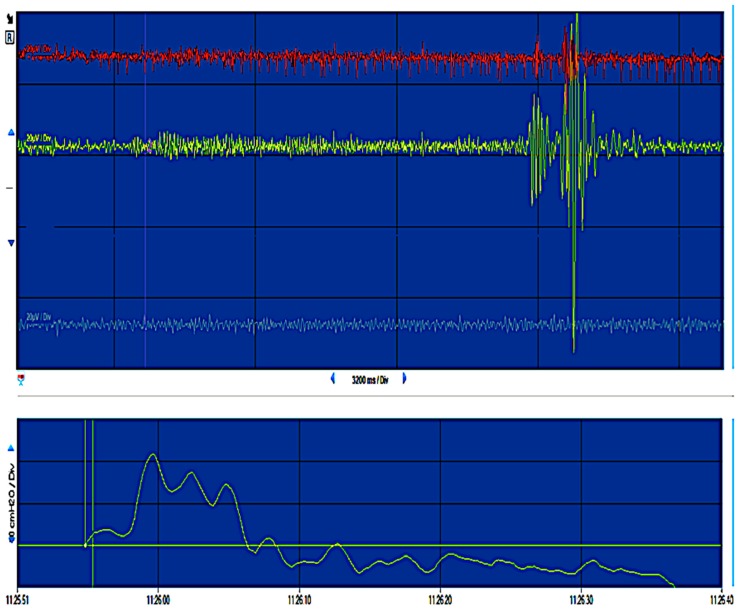
The normal functional reaction after electro-stimulation of the autonomic pelvic nervous system during intra-operative pelvic neuromonitoring after nerve-sparing radical hysterectomy (an example of neuromonitoring report from this study).

**Table 1 cancers-12-00483-t001:** Patient and tumor characteristics.

Characteristics		All Patients *n* = 42 (%)
Age (Years)		Median 45.26 (29–69)
Pre-operative MRI	performed	35 (83.3%)
	Suspected infiltration	10 (23.8%)
	Suspected Lymph nodes	7 (16.7%)
FIGO classification	IA	0
	IB1	5 (11.9%)
	IB2	11 (26.2%)
	IB3	16 (38%)
	IIA1	2 (4.76%)
	IIA2	3 (7.1%)
	IIB	3 (7.1%)
	IIIA	1 (2.38%)
	IIIB	1 (2.38%)
Histology	Squamous cell	33 (78.6%)
	Adenocarcinoma	9 (21.4%)
Grading	G1	0
	G2	21 (50%)
	G3	21 (50%)
Tumour size		44.1 mm (12–100 mm)
TNM classification	IA	0
	IB1	5 (11.9%)
	IB2	6 (14.2%)
	IB3	4 (9.5%)
	IIA1	2 (4.76%)
	IIA2	4 (9.5%)
	IIB	20 (47.62%)
	IIIA	0
	IIIB	1 (2.38%)
Lymph vascular space invasion	L1	23 (54.76%)
	L0	17 (40.47%)
	Missing	2 (4.76%)

**Table 2 cancers-12-00483-t002:** Operation characteristics.

Characteristics		All Patients *n* = 42 (%)
Operation Duration (Minutes) *n* = 35		Median 280 (180–458)
Operative approach	Laparoscopy	24 (57.1%)
	Open surgery	18 (42.9%)
Nerve-sparing technique	bilateral	40 (95.23%)
	unilateral	2 (4.76%)
Lymph node resection	Only sentinel	2 (4.76%)
	Pelvic	25 (59.5%)
	Pelvic and paraaortal	15 (35.7%)
Median number of resected pelvic lymph nodes		44.9 (12–89)
Median number of resected paraaortic lymph nodes		16.7 (5–60)
Cases with affected pelvic lymph nodes (range) *n* = 40		20 (47.6%) (1–45)
Cases with affected paraaortic lymph nodes (range) *n* = 15		4 (9.5%) (1–16)
Parametrial infiltration		22 (52.4%)
Vaginal infiltration		11 (26.2%)
Hospital stay (days) median (range)		10.8 (4–19)
Estimated blood loss (ml) median (range) *n* = 22		97.7 (50–450) ml
Complications (Grade according to Clavien–Dindo classification	Ureter necrosis (GIII)	2 (4.76%)
	Urinary tract infections (GII)	3 (7.14%)
	Sub-ileus (GII)	1 (2.38%)
	Lymphatic oedema (GII)	3 (7.14%)
	Wound dehiscence (GIII)	1 (2.38%)
	Numbness the upper thigh (GI)	3 (7.14%)
	Nausea and vomiting (GI)	5 (11.9%)
Adjuvant therapy	Radiotherapy	Advised 2 (4.76%) Done 1 (2.38%)
	Radiochemotherapy	Advised 21 (50%) Done 17 (40.47%)
	Chemotherapy	1 (2.38%)
Bladder function	Complete healing according to protocol	38 (90.5%)
	Prolonged healing for 2 weeks	3 (7.14%)
	Persistence of impaired sensation	1 (2.38%)
Recurrences	Paraaortal relapse	1 (2.38%)
	Pelvic sidewall	2 (4.76%)
	Inguinal relapses	1 (2.38%)
	Vaginal vault	0
	All recurrences	4 (9.5%)

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
