# Peer review of "Nerve Sparing Radical Hysterectomy: Short-Term Oncologic, Surgical, and Functional Outcomes"

_cancers, 2020, doi:10.3390/cancers12020483_

Round 1

Reviewer 1 Report

I read with great interest the Manuscript titled “Nerve sparing radical hysterectomy: Short-term Oncologic, surgical and Functional Outcomes” (cancers-689122), which falls within the aim of Cancers.    

In my honest opinion, the topic is interesting enough to attract the readers’ attention. Nevertheless, the authors should clarify some methodological points and improve the discussion citing relevant and novel key articles about the topic.

Authors should consider the following recommendations:

The manuscript should be further revised by a native English speaker. Inclusion/exclusion criteria should be better clarified. The Authors did not mention the sample size calculation for their study. It is essential to specify this data in order to guarantee an adequate significance of the results obtained by the Authors. The authors have not adequately highlighted the strengths and limitations of their study. I suggest better specifying these points. What are the actual clinical implications of this study? it is important to report the results obtained by the authors in the context of clinical practice and to adequately highlight what contribution this study adds to the literature already existing on the topic and to future study perspectives. Does this manuscript conform the Enhancing the QUAlity and Transparency Of health Research (EQUATOR) network guidelines? It would be mandatory to declare this element. I could not find any information regarding the approval of the Institutional Review Board. Did the authors obtain this approval before the study start? I suggest highlighting the possibility to perform nerve-sparing and vascular sparing approach (PMID: 27579309; PMID: 31518711) also in case of deep infiltrating endometriosis, in order to save autonomic functions. Accumulating evidence suggests that the management of gynecological cancer should be personalized taking into account the performance status of the patient, in particular in the case of elderly women. It would be interesting to discuss this point of paramount importance, referring to PMID: 30542793; PMID: 29514738.

Author Response

Reviewer 1:

Comments and Suggestions for Authors

I read with great interest the Manuscript titled “Nerve sparing radical hysterectomy: Short-term Oncologic, surgical and Functional Outcomes” (cancers-689122), which falls within the aim of Cancers.    

In my honest opinion, the topic is interesting enough to attract the readers’ attention. Nevertheless, the authors should clarify some methodological points and improve the discussion citing relevant and novel key articles about the topic.

Authors should consider the following recommendations:

The manuscript should be further revised by a native English speaker.

A native speaker revised the text.

Inclusion/exclusion criteria should be better clarified.

This study is a descriptive study. We included all patients with primary cervical cancer, who underwent primary nerve-sparing radical hysterectomy in Charité Medical University of Berlin between January 2017 and June 2019.

The Authors did not mention the sample size calculation for their study.

The study included 42 patients. This was mentioned in the beginning of patients and methods

The authors have not adequately highlighted the strengths and limitations of their study.

We add this paragraph at the end of our discussion:

The strengths of our study are using a standardized operation technique with very clear described surgical steps, all operations were performed by one surgeon to avoid the surgical bias and the clear defined functional tests. A limitation of our study is the retrospective character of the analysis of a prospective collected database.

What are the actual clinical implications of this study? it is important to report the results obtained by the authors in the context of clinical practice and to adequately highlight what contribution this study adds to the literature already existing on the topic and to future study perspectives.

We clearly mentioned that in our conclusion:

Our nerve-sparing radical hysterectomy approach is feasible and applicable even in locally advanced tumors, with good functional results and 90% disease free survival at 18 months of follow-up. These promising outcomes suggest more research and longer follow-up to be able to verify our initial results before being recommended as a reasonable alternative to primary radiochemotherapy as standard of care for locally advanced cervical tumors.

This paper is the first that describes a precise three-dimensional anatomy of paracolpium and its close adherent to the components of pelvic autonomic nervous system. This may be the key for the right adjusting of radicality in cancers limited to the cervix only.

Does this manuscript conform the Enhancing the QUAlity and Transparency Of health Research (EQUATOR) network guidelines? It would be mandatory to declare this element.

This study has been instituted according to template for intervention description and replication (TIDieR) checklist and guide. We added that even in our methods

I could not find any information regarding the approval of the Institutional Review Board. Did the authors obtain this approval before the study start?

Institutional Review Board approval was obtained under registration number EA1/174/14.

We added it in text

I suggest highlighting the possibility to perform nerve-sparing and vascular sparing approach (PMID: 27579309; PMID: 31518711) also in case of deep infiltrating endometriosis, in order to save autonomic functions. Accumulating evidence suggests that the management of gynecological cancer should be personalized taking into account the performance status of the patient, in particular in the case of elderly women.

This good advice could be applied for our next studies. This paper concerned only on cervical cancer patients. The median age of our study collective was 45.26 years (29-69) and all of them had a very good performance status.

It would be interesting to discuss this point of paramount importance, referring to PMID: 30542793; PMID: 29514738. 

We referred to it in the last sentence of our conclusion.

Author Response

Comments

Patients and Methods

page 2  line 50: “Forty-two patients with primary cervical cancer who underwent nerve-sparing radical hysterectomy (type C1) at our institution between January 2017 and June 2019 were included in this  analysis..” The time frame is very short to be able to draw any meaningful conclusion on oncological outcome and the Authors should stess it from the beginning moreover it is not stated  whether this is a prospective or retrospective study.

The study is a retrospective analysis of prospective collected database. We added this sentence at the end of methods.

page 2  line 58: “Therefore, the choice of treatment was based on.” the sentence is incomplete and anyways it suggest significant selection bias in the treatment choice, moreover the Author should have collected data of those patients who declined the surgical approach and received standard of treatment.

We deleted this sentence. The study is not a prospective randomized study to compare the both ways of therapy but a descriptive and observational study

page 2  line 67: “For visual illustration of the autonomic nervous system we used ultrasonic liposuction” The Authors should specify whether such a technique was used during cadaver dissection, confirming that their visual illustrations were obtained at that time, moreover if that was the case the Authors should comment on how to translate such a technique in vivo to help in the identification of the nerves.

The study described the operation technique in Vivo (the described study collective).

page 2  line 78: “The catheter was removed on the third day after laparoscopic nerve-sparing radical hysterectomy and on the fifth day after open surgery.” The Authors should explain why they  removed the catheter at a different interval for the same procedure only based on the surgical approach, knowing that such a choice introduce another bias on the interpretation of functional results

Because the patients will be able to move without help at the third day after laparoscopic surgery, but first at the fifth day after open surgery.

page 2  line 79: “Directly after the surgery, all patients received MYOCHOLINE-GLENWOOD® 10 mg, three times a day for three weeks.” The Authors should give explanations about their choice with references of the literature and comments on the expected  impact on the functional results of such a medical treatment

Bethanechol chloride, a cholinergic agent, is a synthetic ester that is structurally similar to acetylcholine. This is used by urologists for treatment of acute postoperative or postpartum non-obstructive (functional) urinary retention and for neurogenic atony of the bladder with retention (FDA approved drug for this indication). This drug works by neurologically stimulating “muscarinic cholinergic receptors” in the autonomic nervous system. It increases the tone of the detrusor muscle, usually producing a contraction sufficiently strong to initiate micturition andemptying of the bladder. We used this drug to prevent the acute postoperative urinary retention following the surgical dissection on the inferior hypogastric plexus.

Results:

page 9  line 162: “Pre-operative magnetic resonance imaging was performed in 83.3% of patients and revealed suspected pelvic lymph nodes  in 16.7% of cases and suspected infiltration of parametrium in 23.8%” The Authors should explain whether they performed any further investigation in case of suspected pelvic lymph nodes and whether in the case of highly suspected lymph node they would, anyways, carried out the surgery.

The positive lymph nodes are not contra indication for surgery in our institution. We explaind that in the discussion (Of all controlled randomized studies in high-risk cervical cancers, the GOG-109 trial with adding radiochemotherapy after radical hysterectomy yielded the best survival results.)

page 9  line 167: “50% of tumors were bad differentiated”  The Authors should change bad with poorly

We changed it in text.

page 10  line 179 “..if the sentinel lymph node is positive..” the Author should specify whether frozen section of sentinel nodes was always performed

We performed always SLN. We explained it in text (our standard indicates the pelvic and paraaortic lymph node dissection if the sentinel lymph node is positive or if we have suspected lymph nodes in pre-operative imaging).

page 10  line 189 “ The median duration of hospital stay was 10.8 days.” the Author should justify such a long hospital stay

The German health system does not allow shorter hospital stay for such surgery.

page 10  line 198 “In 24 cases, our tumor board advised an adjuvant therapy: in 21 cases (50% of all patients)  concurrent radiochemotherapy, in two patients only adjuvant radiotherapy and in one patient only  adjuvant chemotherapy. Five patients did not follow the recommendation of adding adjuvant therapy..” the Author should define their criteria for adjuvant treatment.

These are the standard criteria from the GOG-109 trial and National Comprehensive Cancer Network (NCCN) Guidelines

Discussion:

page 12  line 244: “These good functional results did  not impair in the 38% of patients who received additional radio- or radiochemotherapie perhaps  because we could achieve a free margins in the entire patient collective, that the adjuvant  radiotherapy was applied in only 45 Gy and did not include a vaginal brachytherapy boost”. This statement seems more a speculation than a real explanation since it is not based on actual data.

This statement depends on the results of GOG 109 trial and the recommendation of NCCN-guidelines:

Postoperative pelvic EBRT with concurrent platinum-containing chemotherapy (category 1)148 with (or without) vaginal brachytherapy is recommended for patients with positive pelvic nodes, positive surgical margin, and/or positive parametrium; these patients are considered to have “high-risk” disease. Vaginal brachytherapy may be a useful boost for those with positive vaginal mucosal margins. Adjuvant concurrent chemoradiation significantly improves overall survival for patients with high-risk, early-stage disease (those with positive pelvic nodes, parametrial extension, and/or positive margins) who undergo radical hysterectomy and pelvic lymphadenectomy.

Posthysterectomy Adjuvant Radiation Therapy

Following primary hysterectomy, the presence of one or more pathologic risk factors may warrant the use of adjuvant radiotherapy. At a minimum, the following should be covered: upper 3 to 4 cm of the vaginal cuff, the parametria, and immediately adjacent nodal basins (such as the external and internal iliacs). For documented nodal metastasis, the superior border of the radiation field should be appropriately increased (as previously described). A dose of 45 to 50 Gy in standard fractionation is generally recommended.

page 12  line 248: “A recurrence rate of less than 10% after a median follow-up of 18 months with locally advanced tumors in two thirds of the cases may be considered as a good result in comparison with  the results of definitive primary radiochemotherapy for the same collective of patients. [15,16]” The comparison between the Authors’ results and the data referred to seems very difficult in terms of significantly different groups, different follow-up and selection bia. The Authors should comment.

We changed the sentence to be: A recurrence rate of less than 10% after a median follow-up of 18 months with locally advanced tumors in two thirds of the cases may be considered as a good result in comparison with the results of definitive primary radiochemotherapy. [15,16] taking into account that the comparison seems here very difficult in terms of significantly different groups and different follow-up.

In the Discussion the Authors should comment on their finding that all the patients who recurred were  nodal positive patients and all of them received adjuvant radiochemotherapy, should be this a group of patients where a different therapeutic approach could be attempted?

We can not draw conclusions from only 4 recurrences, therefor we suggest more research and longer follow-up to be able to verify our initial results

References:

Page 13 line 311: “Muallem MZ, Diab Y, Sehouli J, et al. Int J Gynecol Cancer 2019; 29:1203-1208”. The title of the paper is lacking

We corrected it.

Overall comments:

The Author succeeded in their aim to describe the nerve-sparing radical hysterectomy with  a precise three-dimensional anatomy of paracolpium and of  the components of pelvic autonomic nervous system and in confirming the complete recovery of bladder function in > 90% of patients.

However there are a few limitations partially addressed by the Authors (page 13 line 280 “These promising outcomes suggest more research and longer follow-up to be able to  verify our initial results before being recommended as a reasonable alternative to primary radiochemotherapy as standard of care for locally advanced cervical tumors.”):

Significant selection bias Poor study design and lack of statistical power Very shot follow-up Lack of a control group

Such limitations should be better commented in the Discussion of the study to underline that a meaningful conclusion cannot be drawn from the data presented.

We added this sentence in the discussion:

The strengths of our study are using a standardized operation technique with very clear described surgical steps, all operations were performed by one surgeon to avoid the surgical bias and the clear defined functional tests. A limitation of our study is the retrospective character of the analysis of a prospective collected database and the short follow-up period.

Reviewer 3 Report

see attached file

Author Response

Many thanks for excellent reviewing.

Reviewer 3’s comments and suggestions:

Overall, excellent article, outstanding iconography. Fully deserves publication. The comments of the reviewer are not critical, only given with the intention to improve the manuscript.

Abstract line 3: the word “adherent” should be replaced by “adherence” or preferably “anatomical relationship”.

We changed the word.

 Introduction section section 1. The authors suggest that the general recommendation regarding the “old” stage IB1 (see comment 2) is type C. This does not reflect the attempts at stratifying the risk and consequently the extent of surgery as reported in the ESGO guidelines (Cibula et al. 2019)

We mentioned that (Querleu et al. emphasize the appropriation of the nerve-sparing technique to radical hysterectomy type C adapted to the International Federation of Gynecology and Obstetrics (FIGO) stage IB1 (the old version [3]) with deep stromal invasion). This matches the statement of the ESGO guidelines 2019 (see the figure 3 and 4 in the ESGO guidlines).

National Comprehensive Cancer Network (NCCN) Guidelines 2019 recommends a radical hysterectomy for the new 2018 FIGO stage IB1 (<2 cm), IB2 (2–4 cm), and stage IIA1, and selected stage IB3 (≥4 cm) and IIA2 disease.

In general, the reviewer advises not to use the wording “old” or “new” when referring to the FIGO classification. Dates are required. It might be best to harmonize and use only the 2018- 2019 classification as rightly done later in the manuscript. However, TNM is also mentioned in the text. Consistency is necessary.

We added 2008 to the old version to make it clearer: the old version from 2008

The last sentence of the first paragraph is not clear (“the authors” should be replaced by “some investigators”), and has an aggressive tone (“do not understand” might be replaced by “sadly ignore the possibilities of surgery in more advanced tumors”)

We changed this sentence to: (Some investigators, who claimed that nerve-sparing radical hysterectomy has to be restricted only to small tumors in early-stage cervical cancers; ignored that the technique was designed to deal with vesicovaginal ligament (ventral paracolpium) with sparing the bladder branches of inferior hypogastric plexus. Resecting of ventral paracolpium (vesicovaginal ligament) will be needed only in more advanced tumors.)

“try to” can be deleted from “We try to describe”; the sentence should be rewritten (the section starting with “that” deserves a separate sentence)

We changed it according to your advices.

(In this study, we describe our technique for nerve-sparing radical hysterectomy. This technique incorporates illustrating all parts of the pelvic autonomic nervous system and transecting only the uterine branches in order to preserve the hypogastric nerves, the pelvic splanchnic nerves and the bladder branches of inferior hypogastric plexus. We also report the urologic functional and short-term oncologic outcomes of this technique.)

Patients and method

section 1. Last sentence of the first paragraph is defective

We delete the defective sentence.

Until the publication of (not “publishing”)

We changed the sentence.

4th paragraph: intraoperatively (not “intraoperative”)

We corrected the mistake.

Please specify the rationale and evidence supporting the use of myocholine

Bethanechol chloride, a cholinergic agent, is a synthetic ester that is structurally similar to acetylcholine. This is used by urologists for treatment of acute postoperative or postpartum non-obstructive (functional) urinary retention and for neurogenic atony of the bladder with retention (FDA approved drug for this indication). This drug works by neurologically stimulating “muscarinic cholinergic receptors” in the autonomic nervous system. It increases the tone of the detrusor muscle, usually producing a contraction sufficiently strong to initiate micturition andemptying of the bladder. We used this drug to prevent the acute postoperative urinary retention following the surgical dissection on the inferior hypogastric plexus.

Figure 2 clearly show the medial pararectal, also called Okabayashi, space. Could be pointed

We pointed Okabayashi space in fiqure3.

In the technical description, the preterit tense is used. Should be replaced by present tense

We did not use these words in our text?!

In general, the use of “we” is discouraged. “The nerve is identified” is better that “we identify the nerve”.

We changed it as recommended.

Figures are not called in the text.

We added them in the text.

The angle of view of figure 5 is different from the others. Can be confusing if not mentioned in the legend.

Actually, we use here the same angle of view as in all other figures (cranial and dorsal position of the camera!)

Are the data shown on figure 11 taken routinely (in this case, results should be reported), or is this an example taken in one specifif case? An explanatory legend, or a corresponding explanation in the text would be useful.

This is an example of routinely collected data using pelvic neuromonitoring after nerve sparing surgery. We added this explanation in the legend of figure 11.

 Discussion

 section 1. Fujii et al described in 2017.

We corrected it.

The term “paracolpium” is perfectly adapted. It might be useful to specify the components of it (ventral = vesicovaginal ligament = the akward Fujii’s “posterior leaf of the vesicouterine 2 ligament”, dorsal = rectovaginal ligament, lateral = caudal part of the Nomina Anatomica paracervix).

This point will be discussed in detail in a separate publication about the precise anatomy of paracolpium and why we have to nominate it in this new way.

References 1. The format, reporting or not the issue number, is not consistent.

We corrected it.

Reference 3: last page missing.

We added it

Reference 8 is incomplete.

We completed it.